# Implications of Fragment-Based Drug Discovery in Tuberculosis and HIV

**DOI:** 10.3390/ph15111415

**Published:** 2022-11-15

**Authors:** Mohan Krishna Mallakuntla, Namdev S. Togre, Destiny B. Santos, Sangeeta Tiwari

**Affiliations:** Department of Biological Sciences & Border Biomedical Research Centre, University of Texas at El Paso, El Paso, TX 79968, USA

**Keywords:** fragment-based drug design, drug-discovery, structure-based drug design, mycobacterium tuberculosis, HIV

## Abstract

Tuberculosis (TB) remains a global health problem and the emergence of HIV has further worsened it. Long chemotherapy and the emergence of drug-resistance strains of *Mycobacterium tuberculosis* as well as HIV has aggravated the problem. This demands urgent the need to develop new anti-tuberculosis and antiretrovirals to treat TB and HIV. The lack of diversity in drugs designed using traditional approaches is a major disadvantage and limits the treatment options. Therefore, new technologies and approaches are required to solve the current issues and enhance the production of drugs. Interestingly, fragment-based drug discovery (FBDD) has gained an advantage over high-throughput screenings as FBDD has enabled rapid and efficient progress to develop potent small molecule compounds that specifically bind to the target. Several potent inhibitor compounds of various targets have been developed using FBDD approach and some of them are under progression to clinical trials. In this review, we emphasize some of the important targets of mycobacteria and HIV. We also discussed about the target-based druggable molecules that are identified using the FBDD approach, use of these druggable molecules to identify novel binding sites on the target and assays used to evaluate inhibitory activities of these identified druggable molecules on the biological activity of the targets.

## 1. Introduction

Tuberculosis (TB) is a contagious disease caused by the pathogen Mycobacterium tuberculosis (Mtb) that infects the lungs and can be disseminated to other parts of the body. TB is one of the major causes of death worldwide leading to 10 million cases and ~1.3 million deaths globally [1]. About 8% of the TB incident cases occur worldwide among people living with HIV (PLHIV) [1]. Moreover, 38.4 million individuals were recorded with HIV in 2021 worldwide. HIV infection, which is a major risk factor for TB [2,3], has synergetic effect with TB where the TB accelerates HIV replication and progression [4]. Besides, HIV lowers the immunity and increases the risk of recurrent TB due to the endogenous reactivation or exogenous reinfection [5,6].

TB and HIV are a deadly combination. However, the early diagnosis of both can reduce death from co-infection. In PLHIV, isoniazid preventive therapy and early antiretroviral therapy (ART) can control the infection, whereas using the cotrimoxazole preventive therapy and ART is the best treatment method for people diagnosed or suspected to have TB with an HIV coinfection or infected with TB alone [7]. Another serious concern is the development of drug resistance in TB and HIV. Other than the adverse effects of existing TB drugs and patient incompliance, one of the major causes of development of drug resistance in Mtb is the presence of persister sub-populations. These persisters are highly tolerant to drugs leading to long TB chemotherapy. These persisters might be one of the reasons for the emergence of multidrug-resistant TB (MDR) strains resistant to Isoniazid and Rifampicin, and extensively drug resistance TB (XDR-TB) as MDR-TB plus, which is resistant to the fluoroquinolone drug along with Isoniazid and Rifampicin. As per recent evidence, the mortality rates in patients with MDR or XDR-TB and HIV have been extremely high [8]. Alarmingly, these resistant strains have further undermined the efficiency of current treatment. Therefore, it urgently demands the identification of novel TB and HIV targets as well as compounds against these targets. The identification of essential targets required for the survival or persistence of the pathogen and the new approaches to identify compounds with novel chemical scaffolds, unique characteristics with their mechanism of action that will shorten the treatment and inhibit development of drug resistance. 

In recent times, the fragment-based drug discovery (FBDD), an established strategy to detect small fragments that bind to a specific target. Overall thirty FBDD compounds are already in clinical trials including vemurafenib, kisqali, balversa and venetoclax [9]) for cancer and for TB, NCT03590600, NCT03334734, NCT03199339 and NCT03678688 clinical trials are ongoing for compounds BTZ043, PBTZ169 (macozinone), AZ7371 and OPC-167832, respectively (Figure 1). The FBDD approach consists of several steps including library screening against targets using biophysical, biochemical, and structural biology methods, hit identification, and fragment linking (Figure 2). Recently, a published article from our group described a detailed methodology of FBDD and elaborated current success and challenges of FBDD on mycobacterium [10]. In addition, several groups have been focusing on the current FBDD approach and succeeded in identifying novel molecules for the development of TB and HIV drugs. In this review, we focused on some of the novel fragment hit molecules identified via FBDD and their respective targets, DprE1, KasA, DAPA, EthR, protein antigen 85C, HIV-1 reverse transcriptase, HIV-1 integrase, HIV-1 protease, gp120, gp41, CCR5, and TAR–Tat. Furthermore, we have also summarized the assays used for evaluating the efficacy of these identified fragment hit molecules to inhibit the biological activity of the target molecules.

## 2. TB/HIV Targets

Despite the availability of ART therapy against HIV and six-month 4-drug regimen chemotherapy against drug-susceptible Mtb, both the diseases remain at the top list with high mortality rates worldwide. Additionally, the emergence of drug resistance (DR) and multi-drug resistance (MDR) strains demands improved drug treatments. Having potent compounds against TB and HIV help in controlling these pathogens as HIV patients are highly susceptible to the development of active TB disease either due to infection or reactivation of latent TB. The identification of host factors that can control both the pathogens TB and HIV will revolutionize the field; however, so far, no such targets have been identified and no studies on FBDD fragments against such host-directed targets have been performed. The FBDD approach has been successfully explored on the disease targets by several studies, and in this review, we are focusing on some of the targets of TB/HIV.

### 2.1. FBDD Fragment Hit Compounds against Mycobacterium Tuberculosis Targets

The FBDD approach has been successfully explored by several studies, which we have summarized in our earlier review [10]. In the present section, we have focused on the few Mtb target molecules, which have gone through in vivo animal studies or clinical evaluation. As most of the existing drugs that are in use in the clinic or clinical trials target Mtb cell wall or bioenergetics pathways, in this review we have also discussed the updated status of the novel biosynthesis pathways as biotin and arginine as novel target molecules shown to be important for Mtb survival and persistence.

#### 2.1.1. Decaprenylphosphoryl-β-D-ribofuranose 2′-oxidase 

Several enzymes involved in metabolic pathways in the Mtb are being explored as potential targets for anti-Tb drug discovery. One such enzyme is decaprenylphosphoryl-β-D-ribose 2′-epimerase (DprE1), which is heteromeric and encoded by dprE1 (Rv3790) and dprE2 (Rv3791) genes, respectively [11]. DprE1 is involved in the metabolic pathway through which mycobacteria produce an arabinogalactan scaffold, which is an integral part of the Mtb cell wall structure [12]. Several studies have reported that the absence or reduced expression of functional DprE1 makes Mtb non-viable [13,14].

DprE1 has been successfully explored by several studies in search of small molecule inhibitors against it [15,16,17,18,19,20,21,22,23,24,25,26,27,28,29]. Recently, inhibitor molecules such as NCT03590600 (BTZ043), NCT03334734 (PBTZ169), NCT03199339 (AZ7371) and NCT03678688 (OPC-167832) have reached a clinical evaluation state against Mtb DprE1. A recent study by Borthwick et al. (2020) identified a series of morpholino-pyrimidine DprE1 inhibitors. In the initial screening using SAR, authors have found piperidinylpyrimidine derivatives with minimum inhibitory concentrations (MIC90) of H37Rv that were 30.6 μM and 15.6 μM. The synthesis of a series of compounds and the replacement of the distal piperidine ring, benzyl group, and pyrimidine core in search of potent inhibitor molecules have been reported by the authors. This has led to the discovery of compounds that have to promise in vivo efficacy against acute Mtb infection in murine mice models [12].

#### 2.1.2. KasA

KasA is another essential enzyme involved in the mycolic acid synthesis pathway. In earlier studies with isolated natural products, the antibiotic sensitivity of the KasA enzymes was highlighted, paving the way for anti-Tb drug discovery [30,31]. Kapilashrami et al. (2013) synthesized thiolactomycin (TLM) and pantetheine analog, PK940, using the Interligand NOEs and FBDD approach. The kinetic binding data and SAR studies revealed time-dependent inhibition in this KasA enzyme system. The authors also reported that the inhibitors potency was increased due to the elaboration of the thiolactone ring at C3 using a fragment-growing approach. The subsequent modification of the methyl group with larger groups resulted in increased binding affinities. However, this led to a significant decrease in ligand efficiency, which can be retained by further chemical modification [32].

#### 2.1.3. EthR Transcriptional Repressor (EthR)

The mycolic acid synthesis pathway is another interesting target for anti-Tb drug discovery. Conventional drugs such as isoniazid and ethionamide target the same component involved in this, resulting in hampered mycolic acid production and hence MTB growth. However, the concentration of ethionamide currently used causes liver toxicity and hence several approaches are currently being explored to lower this concentration. Both drugs are pro-drugs and are shown to have different resistance profiles as different bacterial enzymes activate them [33]. The EthA activates ethionamide and the ethA gene synthesizes it [34]. The transcriptional repressor EthR regulates the ethA gene [35]. Therefore, current drug discovery research focuses more on EthR inhibitors, which can serve as adjuvants for ethionamide.

Based on pharmacophore modeling, researchers from the Pasteur Institute found several hits [36]. BDM14500 was the first inhibitor of EthR that was identified through the library screening. In further search of new potent EthR ligand molecules to treat MDR-TB, Villemagne et al. (2014) identified potent EthR ligands that enhance the in vitro and in vivo ethionamide activity. They selected two series of compounds (BDM31369 and BDM31827) and successfully co-crystallized with EthR. These two ligands occupy the ligand binding domain of EthR and both interacted to Asn179 (Asparagine). Later, in different library screens, the same authors identified compound **3** (4-Iodo-N-prop-2-ynylbenzenesulfonamide) that was considered as a fragment molecule with <3 H bond acceptor and H bond donor atoms, low molecular weight, and solubility for in silico analysis. Followed by fragment-based hit-to-lead optimization, this fragment sulfonamide group shared a critical hydrogen bond interaction via Asn179 in the binding pocket of EthR [37]. Based on the binding mode, the authors have synthesized a series of compounds. The resulting SAR data with 32-fold improved inhibition of EthR DNA binding were found with compound **18** (4-Iodo-N-prop-2-ynylbenzenesulfonamide derivative) benzenesulfonamide). Additionally, these results were confirmed ex vivo and they found a boost in the antibacterial effect of ethionamide at 1/10 its MIC (MIC99 μg/ml): BTZ043: 0.001; PBTZ169:0.0003). Additionally, authors have used crystallography to validate the binding mode of the compound to the predicted one [33]. Another study by Surade and colleagues used thermal shift analysis (TSA) to screen a library of 1250 compounds and found 86 hits, some of which could stabilize EthR by more than 5 °C. Following validation using orthogonal biophysical techniques and primary and secondary screening, the SPR-validated hits were soaked into EthR crystals to elucidate their binding modes. The authors reported another hit 21 (4-Iodo-N-prop-2-ynylbenzenesulfonamide derivative) and it was more interesting because of its ability to bind both the second subpocket in the crystals and the hydrophobic cavity simultaneously. The linking of slightly modified compounds resulted in the formation of compound **22** (4-Iodo-N-prop-2-ynylbenzenesulfonamide derivative) with a decreased LC50. However, the observed identical in vitro antibacterial potencies of these compounds highlight the ineffectiveness of the straightforward optimization of binding strength and LE. Furthermore, Nikiforov et al. (2016) suggested that fragment merging solely based on biophysical techniques was found to be unsuccessful [33,38]. Instead of a biophysical approach, structure–activity relationships (SAR) exploration of compound **20** 4-Iodo-N-(1-{2-oxo-2-[4-(3-thiophen-2-yl-1,2,4-oxadiazol-5-yl)-piperidin-1-yl]-ethyl}-1H-1,2,3-triazol-4-ylmethyl)-benzenesulfonamide yielded submicromolar ethionamide boosters such as 23 (4-Iodo-N-prop-2-ynylbenzenesulfonamide derivative) [39].

Another study by Flipo et al. (2012) used a novel whole mycobacteria phenotypic assay to screen the library of 14640-compounds and found out compound **1** (N-phenyl-phenoxy acetamide derivatives) binds with EthR which was further confirmed by an X-ray diffraction study. Additionally, the use of TSA followed by in vitro testing against Mtb-infected human macrophages led to the discovery of two potent N-(4-benzothiazol-2-ylphenyl)-2-phenoxy acetamide derivatives. Compared to compound **1**, compounds **4** and **5** were 50-fold more active in boosting ethionamide subs active dose on Mtb-infected macrophages [40].

#### 2.1.4. Antigen 85C

Mycolic acids are the major and essential components of the Mtb cell wall (47). The Antigen 85 (Ag85) complex is constituted of three different enzymes that transfer trehalose monomycolate (TMM) to arabinogalactan or another TMM to form ′-trehalose dimycolate (TDM) [41,42,43]. In Mtb, earlier mutation studies have shown that the mutant lacking Ag85C had altered cell wall permeability with a 40% decrease in cell-wall-linked mycolic acids. As they have a ~70% sequence similarity, they can be targeted by a single molecule. Scheich et al. (2010) used the FBDD approach based on 15N-HSQC NMR spectroscopy and screened a diverse chemical library of 5000 compounds. The authors reported that one of the six initially identified fragments had shown a MIC of 50–100 μg ml^−1^ against M. smegmatis and MIC of 500 μg ml^−1^ against Mtb H37Rv and an MDR strain. However, further preparation of crystal structures of Ag85C–inhibitor complexes was not successful. Hence, the authors tested different chemical substitutions in similar analogs to increase the lead potency and found the best compound had a MIC of 20–50 μg ml^−1^ against Mtb H37Rv and an MDR strain [42,44].

#### 2.1.5. 7,8-Diaminopelargonic Acid (DAPA) Synthase (BioA)

The enzymes involved in the biosynthesis of biotin (vitamin H) are explored as targets for antibiotics [45]. Antibiotics such as actithiazic acid [46] and amiclenomycin (ACM) [47,48,49,50,51] isolated from the Streptomyces species have been found effective against mycobacteria and target the enzymes involved in the biotin biosynthesis pathway. However, the effect of ACM and its derivatives as Mtb BioA inhibitors was found to be very low in animal models of Mtb infection [48,52,53]. The biosynthesis of the biotin pathway is another crucial player involved in mycobacterial survival in vitro and in vivo. The gene BioA, which encodes for DAPA has been implicated in the long-term survival of mycobacteria [54]. Park et al. (2011) showed the role of the enzyme BioA in Mtb persistence in a murine model [53]. Further Mtb deletion mutants, ΔbioA, resulted in the inactivation of the enzyme BioA, resulting in cell death and growth arrest, indicating the importance of BioA enzyme targeting in anti-Tb drug discovery [53].

The very first use of FBDD has been shown by a study carried out by Finzel and co-workers in which authors screened a library of 1000 chemical compounds at 5 mM using a thermal shift assay. Among them, 2-(aminomethyl)-benzothiazole 23 was found to mimic the interactions of the natural substrate 7-keto-8-aminopelargonic acid (KAPA), as confirmed by ligand-based STD NMR experiments [55,56]. In addition, the authors used TSA to evaluate commercially available benzothiazoles related to the 2-(aminomethyl)-benzothiazole 23 in structure-guided fragment optimization. Out of these, the hydrazine derivative 24 showed similar bonding patterns and conformational changes in the enzyme-binding site, which was further confirmed by X-ray crystallography. Based on the observed conformational change in BioA induced by these compounds, the authors hypothesized the development of possible inhibitors that can stabilize catalytically inactive forms of the BioA enzyme [55,56].

#### 2.1.6. Arginine Biosynthesis Pathway

Earlier studies have evaluated the possible involvement of the de novo L-arginine biosynthesis pathway having a crucial role in the adaptive response to the oxidative stress generated by vitamin C and INH [57,58]. To the best of our knowledge, the arginine biosynthesis pathway is the only amino acid pathway targeted by any TB drugs. Tiwari et al. (2018) have shown that the in vitro and in vivo sterilization of Mtb ΔargB and ΔargF mutants without the emergence of suppressor mutants occurs with the arginine deprivation of Mtb persister populations. In a recent study from our laboratory using the FBDD approach, we checked the ligand ability of ArgB, ArgC, ArgD, and ArgF enzymes involved in L-arginine biosynthesis in the biosynthetic pathway. Followed by hit identification and validation using biochemical and biophysical assays, we have confirmed the on-target activity of identified compounds against ArgB [58,59]. The conserved nature of the arginine biosynthesis pathway among Mtb and NTMs will be helpful to identify novel inhibitors against the enzymes involved in this pathway [10,60].

In addition to these molecules, several studies have successfully explored the FBDD approach in search of potent inhibitor molecules against several targets such as Inosine-5′-monophosphate Dehydrogenase [61], Protein tyrosine phosphatase [62], Dehydroquinolase [63,64], type-2 NADH dehydrogenases [65], Cytochrome P450 enzymes [56], etc. The structure of the TB targets and its fragment hit molecules are listed in Table 1 and their potency parameters are listed in Table 2.

### 2.2. Human Immunodeficiency Virus Targets

Viruses are infectious microbes surrounded by a protein coat. A virus cannot replicate by itself, instead, it uses host cell components for the generation of viral particles after entering the host. Therefore, the virus genetic material (DNA or RNA) injected into the host cell DNA for replication is one of the important steps in the virus life cycle. Specifically, several essential enzymes including reverse transcriptase, integrase, protease, gp120, gp41, CCR5, and TAR/Tat are involved in the life cycle of HIV. The blockage of these HIV enzymes prevents virus proliferation [71]. These HIV components were identified as a drug target. The inhibition of these targets is the primary goal for the therapy of HIV infection.

#### 2.2.1. HIV-1 Reverse Transcriptase

HIV-1 reverse transcriptase is an essential viral enzyme that catalyzes the single-stranded viral RNA (ssRNA) into double-stranded DNA (dsDNA), which integrates into the host genomic DNA. Reverse transcriptase acts as RNA-dependent, DNA-dependent DNA polymerase activities along with RNase H activity, which is involved in the early stage of the viral life cycle. These reverse transcriptase functions are required to generate a dsDNA copy of the viral RNA genome [72,73]. HIV-1 reverse transcriptase enzyme comprised a heterodimeric subunit, p66 (catalytic), and p51 (structural) [74]. HIV-1 reverse transcriptase is a therapeutic target for FBDD and structural-based drug design that inhibit the replication of HIV. Currently, two types of HIV-1 reverse transcriptase inhibitors are available for clinical use, NRTI (nucleoside/nucleotide analog reverse transcriptase inhibitor) and NNRTIs (non-nucleotide reverse transcriptase inhibitor). Examples of NNRTIs are nevirapine and efavirenz (first-generation NNRTIs) and rilpivirine and etravirine (second-generation NNRTIs) [75]. These NNRTIs show different types of chemical structures than NRTIs. HIV-1 reverse transcriptase is an ideal target for allosteric inhibition due to its flexibility allowing inhibitors to bind to the one binding site that can affect the enzyme at a distant location [76]. That the smaller chemical fragments are enough to act as a probing molecule to identify the allosteric binding sites of a target is an advantage of FBDD. The fragment-based approach has been used to identify novel drugs (NNRTIs type) and search for alternative binding sites on reverse transcriptase that may act as a drug target. A study from Geitmann, M. et al. (2011) identified 96 fragment-sized compounds for the binding of HIV-1 reverse transcriptase using SPR assay. These hits were screened for inhibition of reverse transcriptase polymerase activity and their ability was checked with a known inhibitor, nevirapine. In the final analysis of screening, the authors found one fragment (4-bromo-1-indanone) with a reproducible inhibitory activity against drug-resistant HIV-1 reverse transcriptase variants (K103N, Y181C, and L100I) (IC_50_ < 25 μM) [77]. Bauman and his colleagues detected three novel allosteric druggable sites (NNRTI-adjacent site, knuckles site, and incoming nucleotide-binding site) with one fragment for the inhibition of HIV-1 reverse transcriptase using X-ray crystallography-based FBS [78]. Furthermore, some of the fragment-based NBD (N-phenyl-N-piperidin-4-yl-oxalamide analog designated as NBD) compounds showed moderate activity against HIV-1 reverse transcriptase. For example, the polycyclic NBD compounds inhibited reverse transcriptase with an IC_50_ value between 28 and 45 μM [79]. However, the thiazole-containing scaffolds with amine piperidine ring fragment 17 NBD compounds showed effective antiviral activity against reverse transcriptase with the IC_50_ of 1.7 ± 0.2, where nevirapine used as a positive control with IC_50_ of <0.5 μM [80]. Interestingly, the novel fragment-sized inhibitors were identified for HIV-1 reverse transcriptase using saturation transfer difference (STD) NMR and in vitro enzyme inhibition assay (two complementary methods) that are mechanistically different from NRTIs and NNRTIs. The fragments 4, 5 (oxime), and 8 (P-hydroxyaniline) inhibited RNA- and DNA-dependent DNA polymerase activity of reverse transcriptase in the micromolar range and retained the inhibitory activity against drug resistant reverse transcriptase variants, Y181C, K103N, or G190A. Moreover, fragment 8 displayed the inhibition of HIV-1 reverse transcriptase RNase H activity and also inhibited the HIV-1 replication in TZM-Bl cells [81].

#### 2.2.2. HIV-1 Integrase

HIV-1 integrase is involved in the insertion of dsDNA of the viral genome generated by reverse transcriptase into the host chromosomal DNA [82]. Hence, the new viral particles are generated by host transcription and translation machinery. The integrase enzyme is a tetramer consisting of two active sites and each monomer has three domains that have different functions, i.e., an N-terminal domain (NTD) for protein multimerization, a C-terminal domain (CTD) for non-specific DNA binding, a central core domain (CCD) for DNA substrate recognition, and a catalytic motif (DDE), which is involved in the translocation and stabilizes the integrase-DNA (IN-DNA) complex. The inhibitors currently in clinical use include IN-STIs (integrase standard transfer inhibitors) that specifically bind to the catalytic active site of integrase. Allosteric inhibitors target different sites on the integrase CCD domain, such as ALLINIs (allosteric integrase inhibitors) that bind to the dimer of HIV-1 integrase and interact with a host cell factor called lens epithelial-derived growth factor (LEDGF), essential for integration [83,84,85,86]. The peptide-type inhibitors of LEDGF interaction and multimerization as BI-224436, have also been reported in several studies and represent a new class of non-catalytic site inhibitors (NCINIs) that have entered phase 1 clinical trials for the first time [87,88]. Later, Mitchell et al. (2017) found next generation NCINIs (GS9695 and GS9822) with higher potencies than previous NCINIs. Interestingly, GS9822 showed significantly higher resistance barrier than the approved reverse transcriptase inhibitor rilpivirine [89]. Further development of these NCINIs has stopped because of vacuolation of the bladder urothelium in cynomolgus monkeys but not in rats, however, these symptoms were absent in parent NCINI inhibitor BI-224436 that is in a phase I clinical trial. Further experimental evidences and the development of NCINIs are required to be used as HIV drug molecules [90]. These NCINI inhibitors have been characterized extensively and are in the advanced preclinical development stage [91,92,93].

HIV-1 integrase is an essential target for FBDD strategies. Compared to reverse transcriptase, the structure of integrase consists of large tetrameric protein subunits, hence it poses more structural challenges to identify inhibitors using FBDD. The unavailability of full-length HIV-1 integrase crystal structures further enhances the challenges in rational drug design as it has to depend on the subunit crystal structures of HIV-1 integrase [94,95]. The FBDD studies have been targeting the LEDGF/IN and allosteric binding sites using different FBS methodologies. The Wielens group employed STD-NMR as a primary screen and X-ray crystallography was used for the characterization of the fragment hits. The binding of these fragments was investigated using chemical shift ^15^N HSQC spectra; apparently the number of fragments were bound at novel binding sites on the integrase, termed the fragment binding pocket (FBP), and could achieve a 17-35-fold improvement in the binding affinity by introducing bromine to the compound **14** (N-benzyl indolinone analogs) [96]. Nevertheless, the Rhodes group identified a new site of integrase for the structure-based design of allosteric inhibitors, unlike the previously reported data [97]. One of these hits, an N-benzyl indolinone 15, bound to a novel pocket on HIV-1 integrase. The binding site Y3 is located on the mobile loop of integrase. A fragment hit called Y3a was bound to Y3 and inhibited the HIV-1 integrase with IC_50_ of 259 μM in strand transfer assay and the fragment extended analogs, Y3b, inhibited the integrase with an IC_50_ of 5 μM [97,98]. Similarly, compound **16** showed 45-fold more efficiency than 15 and when bound to the HIV-1 integrase, the central core domain resulted in an increased interaction with Q62 and H183 that defined the binding site [81]. The compounds **17** and **18** and their analogs (N-benzyl indolinone analogs) were investigated using X-ray crystallography, a strand-transfer assay, a cell-based assay, and an alpha screen assay that confirmed that these compounds were specifically bound to the LEDGF site on HIV-1 integrase [99]. The most important and potent compound **19** (N-benzyl indolinone analog) was active in both biochemical assays and against HIV-1 integrase replication in the cell culture with the EC_50_ of the 29 micromolar range [100]. Furthermore, the thiophene of KM00835 and the aromatic ring of SB00942 bound at the FBP site in an overlapping manner leading to the series of benzodioxole compounds from both hits. The best-hit compound in this series is KM-SB3, which had an IC_50_ of 170 μM against HIV-1 integrase [101]. However, there is no further improvement of inhibitor scaffolds or development of novel binding sites from the studies that have been published to date. Overall, the available novel LEDGF inhibitors have provided a better understanding of these effects and could be exploited in the improvement of a novel class of antiretroviral drugs targeting HIV-1 integrase.

#### 2.2.3. HIV-1 Protease

HIV-1 protease, an enzyme responsible for the degradation of Gag-Pol polypeptide into viral proteins, is an important step for viral maturation. HIV-1 protease is a dimeric aspartic protease from the aspartic protease family. After reverse transcriptase and integrase, protease is a primary drug target that has been extensively exploited. HIV-1 protease exhibits broad substrate recognition. The generation of HIV-1 protease inhibitors led to the development of the HAART (Highly Active Anti-Retroviral Therapy) cocktail regime, a combination of HIV-1 reverse transcriptase and HIV-1 protease inhibitors. This HAART cocktail drug significantly extends the life span of HIV-infected patients and exhibits the best dose response [102]; at the same time, protease is prone to develop drug resistance [103]. The available FDA-approved inhibitor drugs in the market showed drug resistance profiles, with an exception of darunavir [104]. To overcome these hurdles, there is an urgent need to find novel drugs with novel mechanisms of action. The fragment-based approach is an attractive method for the identification of suitable binding sites on HIV-1 protease.

A study from Perryman et al. (2010) employed fragment-based screening against HIV-1 protease using an active site fragment library, which contains 384 different chemical fragments, and the library was screened by crystal soaking and crystallization. They identified a novel hit, 3- bromo-2,6-dimethoxy benzoic acid (Br6) in the flap site and 1-bromo-2-naphthoic acid (Br27) in the exosite of HIV-1 protease [105]. However, the exosite of the HIV-1 protease was described as a potential allosteric site [106]. Furthermore, Tiefenbrunn et al. (2014) conducted a crystal screening with IF1 (Indole-6-carboxylic acid) and 2F4 (2-methylcyclohexanol) bound in the flap site of HIV-1 protease and 4D9 (2-methylcyclohexanol) bound in the exo-site of HIV-1 protease [107]. The pitfalls of crystallographic fragment screens can be overcome by a halogenated fragment library that allows unambiguous identification of the halogen binding site. This approach has been successfully applied for fragment screening at SGX pharmaceuticals [108]. The performed crystal soaking with brominated fragments bind at three different binding sites within the protease crystals. A thorough understanding of exo-site binding and their interactions with fragment binding site residues revealed a large surface pocket to be targeted for drug designing. Fragment expansion and elaboration are the next following steps after fragment discovery, along with fragment linking. The result in large molecules is being tested for activity against the target. Several compounds were observed that completely inhibit the nucleation of HIV-1 protease. These compounds were subjected to co-crystallization with HIV-1 protease, confirming that the binding of hit molecule IF1 in cocrystals of apo HIV protease and pepstatin-inhibited HIV protease. However, the inhibition of these compounds was not in the range of 1mM [98]. Therefore, novel techniques are to be identified for the large molecules with a high binding affinity that inhibits HIV-1 protease.

#### 2.2.4. Gp120

Envelope glycoprotein gp120 (Gp120) is a glycoprotein located on the surface of the HIV envelope. Gp120 is essential for the virus’ entry into the host cells and it plays a crucial role in the attachment of host cell receptor CD4 on T cell lymphocytes. The binding of CD4 induces the cascade, leading to conformational changes in gp120 and gp41 (fusion protein) that allow the fusion of the viral membrane with the host cell membrane. The attachment of gp120 is the primary step for the establishment of an HIV infection, hence, gp120 is the initial target for vaccine research and the development of novel inhibitor drugs. The currently licensed drugs in clinical use are maraviroc, which bind to the co-receptor CCR5 and block the binding of gp120 to the host cell, and Fostemsavir (BMS-626529), a small molecule inhibitor, which binds gp120 and prevents viral entry into the host cell. Targeting gp120 is extremely difficult due to its high degree of variability. Another small molecule NBD-556 was the first reported inhibitor against HIV-1 gp120 [109], however, instead of inhibiting the HIV-1 infection it turned out to be a gp120 agonist and lead to the establishment of HIV infection in CD4^−^ CCR5^+^ cells [110,111]. Later, researchers modified NBD-556 to generate various NBD-556-based molecules. Recently, Iusupov et al. (2021) replaced NBD-556 (2,2,6,6-Tetramethylpiperidine) with a (4-methyl-2-(piperidin-2-ylmethyl) thiazol-5-yl) methanol scaffold that resulted in NBD-09027 [80], which showed partial agonist characteristics. Later, using scaffold-hopping approach the same group converted oxalamide of NBD-09027 to a pyrrole transformation leading to synthesis of a new compound “NBD11021”. Interestingly, though the parent compound NBD-09027 is gp120 agonist but the new compound NBD11021 showed complete opposite characteristic as a gp120 antagonist [110]. The gp120 was transformed into an antagonist after the optimization of NBD-556 using a scaffold-hopping approach and exhibited potent anti-HIV-1 activity [80]. Furthermore, they successfully validated the synthetic compounds based on NBD-14136 amine and acid intermediate using isosteric replacements and were able to exhibit antiviral activity and cell toxicity. Among them, compound **12** (Indole based acids) was equipollent to the NBD-14091 and NBD-14092 and showed antiviral activity than compounds **17** and **13** (thiazole-amine moieties) [80].

In addition to these, some of the novel small molecules have been reported; these are called non-natural amino acids termed 882376 and showed HIV-1 inhibitory activity. The performed experiments on mutant viruses (K348R, S411G, K432E, C604Y, and C764R) suggested that 882376 may target the CD4 binding site in gp120 and prevent the HIV-1 infection by disrupting the interaction of gp120/CD4 [111]. Interestingly, N (2-(2-guanidinoacetamido)-1-(4-(hydroxymethyl) thiazol-2-yl-ethyl)-54(trifluoromethyl)phenyl)1 H-pyrrole-2-carboxamide (compound **10**) showed high antiviral activity (infected with pseudovirus HIV-1_HXB-2_) and cytotoxicity in TZM-bl cells among the NBD compounds, where an AWS-I-169 molecule was used as a control. Guanidium compound **10** is also to be considered as a water-soluble novel inhibitor lead for further optimization and is expected to interact with the binding residues Arg59 and Asp368 of the gp120. A moderate result could be achieved with the thiazole ring containing compounds NBD-14010, NBD-14171, and NBD-14136 [112].

#### 2.2.5. Gp41

Gp41 is an envelope glycoprotein involved in viral fusion with the host cell. The transmembrane protein gp41 undergoes a conformational change upon the binding of gp120 to a cellular co-receptor. The gp41 exists as a trimer consisting of three domains, a cytoplasmic domain, a transmembrane domain, and an ectodomain. All these domains have three important functional regions, fusion peptide, N-terminal heptad repeat (NHR), and C-terminal heptad repeat (CHR), which are responsible for the fusion. Enfuvirtide, the first FDA-approved gp41 peptide inhibitor, prevents the fusion of the virus into the host cell. Additionally, NHR directly interacts with membranes and is actively involved in the fusion process, therefore, it is an attractive target for the FBDD-based HIV drug design [113,114]. The NHR subpocket of gp41 is an especially attractive novel small molecule target for the generation of novel inhibitors. In order to identify inhibitors against the NHR subpocket of gp41, Tiefenbrunn et al. (2014) performed an NMR fragment screening experiment and screened a 500-member library and found one potential hit 1a (4-methyl-5-thienyl-3-aminopyrazole), but this fragment failed to bind in the hydrophobic pocket of gp41. However, the authors were able to successfully show the binding of the 1a fragment with gp41 using NMR analysis combined with Autodock Vina calculations. To identify more compounds that bind to the NHR/CHR of gp41 and to further confirm the binding of 1a to the NHR subpocket of gp41, the same authors screened a fragment library and identified 29 fragments using a rapid overlay of chemical structure (ROCS) method. Finally, seventeen fragments were found to bind the NHR/CHR of gp41. These exciting results suggest that the NHR region has a druggable pocket that is located next to the hydrophobic binding pocket that is targeted by non-peptidic gp41 inhibitors [98]. Therefore, a well characterized fragment molecule might lead to a powerful binding inhibitor with high binding affinity and could be achieved using a fragment-linking approach.

#### 2.2.6. CCR5 Co-Receptor

CCR5 is a co-receptor located on the surface of host T-cells that binds to gp120, which initiates the entry of HIV. CCR5 is essential for the infection of HIV. Therefore, blocking CCR5 interaction with gp120 can block HIV infection. Maraviroc is an FDA-approved drug that acts as a CCR5 antagonist. CXCR4 is another co-receptor present on the cell membrane that is involved in increasing the growth of the virus [115]. Some of the HIV-infected patients develop resistance against the drugs via mutations in gp120 or gp41 that allow the HIV to bind to CCR5. Therefore, the search for small molecule inhibitors against CCR5 is needed to treat HIV-resistant strains. Navratilova et al. (2011), identified small molecules that bind to CCR5 using a surface plasmon resonance-based methodology [116]. Five fragment-like hits were observed that bound to the maraviroc binding site of CCR5 [107]. Further development has to be performed on these fragments with higher affinities.

#### 2.2.7. TAR–tat Interaction

The HIV-1 transactivation response (TAR) element is a stem-loop structure present in the pro-viral long terminal repeat (LTR) at the 5’ end of the viral mRNAs that bind the viral Tat protein and the host cyclin T1 protein that triggers the cyclin-dependent kinase 9 (CDK9) recruitment. CDK9 leads to hyperphosphorylation of RNA polymerase II, which enhances transcriptional activity. This step is crucial for viral replication. There are no clinical drugs that have been identified that target HIV-1 TAR, indicating it is a novel target for the inhibitors. However, the Davidson group conducted an NMR-based fragment screen against TAR/tat using a target-specific chemical probe [117]. The MV2003, an arginine-derivative molecule was used as a probe for the fragment screening. This probe resembles the Tat arginine residue and is involved in the binding of TAT. Thereby, the MV2003 probe detects the arginine binding pocket of TAR. These fragments were assessed using NOE spectroscopy for their capacity to bind to either TAR or MV2003 or with both TAR and MV2003 in a solution but none of these fragments were unable to bind individually, except the TAR–MV2003 complex [98]. Researchers further used NMR for the investigation of fragment binding that resulted in these fragments only being bound to the complex of TAR with MV2003 but not alone. Furthermore, the binding studies performed with TAR: MV2003 complex using ^1^H-^1^H NOESY spectra, revealed MV2003 binds at the top of the RNA helix of TAR. The ternary complex of TAR with MV2003 was determined using HADDOCK, a computational docking tool that indicated that these fragments were located within the major groove of the RNA [98]. The TAR–tat interaction is an important target to be explored. We have summarized HIV targets and their known fragment hit molecules with their biological activity in Table 3. Further we have elucidated the structures along with their published hit fragments in the Table 4. Additionally, we have also added potency of these identified fragments in Table 5.

## 3. Conclusions

Long chemotherapy due to persister subpopulations and toxic side effects of the drugs leading to patient incompliance in the treatment of TB and HIV causes the development of drug-resistant strains. The primary goal is to improve the efficacy and shorten the duration of active treatment. Hence, a new type of drug molecule with a high potency, less toxicity, and a new mechanism of action are required to effectively treat and control drug-resistant Mtb and HIV cases and the emergence of drug-resistant pathogens. Additionally, novel preventive strategies and screening methods for the generation of anti-HIV and anti-TB drugs are also needed to combat these diseases. The fragment-based drug discovery offers a new potential platform or approach for the rapid screening of drug-like structures for evaluation and to initiate the probing of a potential suitable target. This approach has been very successful in the identification of novel hit molecules leading to the development of drugs already in use in clinics suggesting it holds potential for drug development for TB and HIV. The ongoing research on TB and HIV using FBDD for the identification of novel fragment hit molecules has identified new inhibitory binding sites or allosteric sites for inhibition on target molecules that are currently in clinical trials. In this review, we have discussed the implications of the FBDD approach that has successfully identified drug-like molecules against TB and HIV-1. Interestingly, identified hits using the FBDD approach against TB and HIV-1 exhibited inhibitory activity on infected cell lines or in vivo in animal models. Therefore, these novel fragment molecules offer the promising potential to act as TB and HIV inhibitors in pre-clinical and clinical evaluations. Furthermore, consideration of novel targets other than those targeted by existing drugs and the potential to eliminate persisters will pave the way to shorten chemotherapy and the emergence of drug resistance in pathogens. Overall, the FBDD-based techniques and its programs play a vital role in contributing to the development of efficacious novel agents to be used in therapeutics.

## Figures and Tables

**Figure 1 pharmaceuticals-15-01415-f001:**
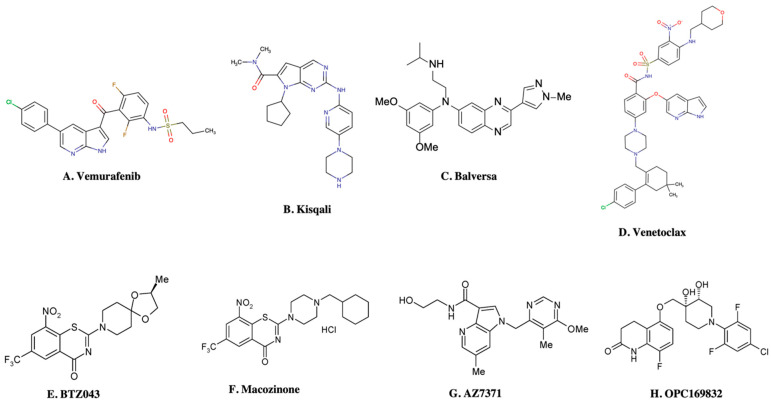
FBDD inhibitors in clinical trials for cancer (**A**–**D**) and TB (**E**–**H**).

**Figure 2 pharmaceuticals-15-01415-f002:**
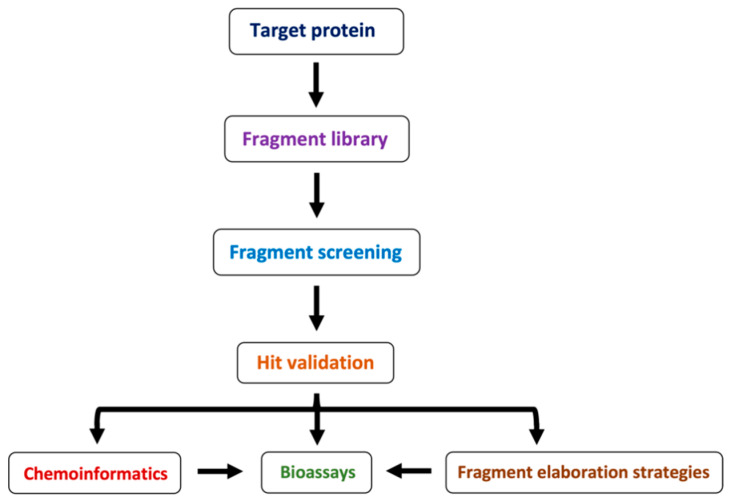
Schematic representation of FBDD methodology.

**Table 1 pharmaceuticals-15-01415-t001:** TB targets and its fragment hit molecules.

Structure	Fragment Hit	Reference
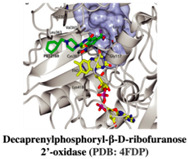	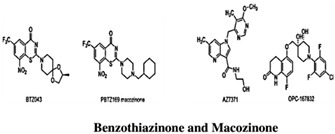	[12,21,66]
* 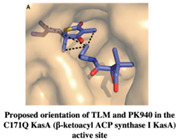 *	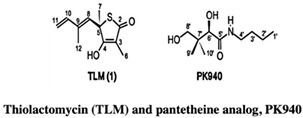	[32,67]
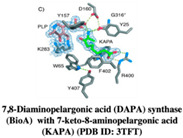	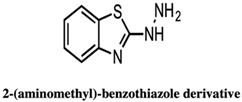	[55,56]
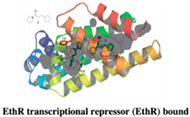	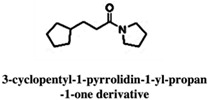	[39]
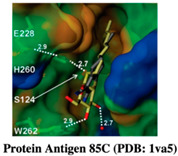	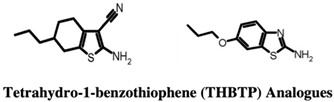	[44]

**Table 2 pharmaceuticals-15-01415-t002:** TB fragment molecules with potency parameters.

S.No.	Target	Fragment Molecule	Potency Parameters	References
1.	DprE1	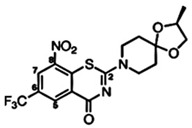 BTZ043	MIC99 1 ng mL^−1^; 2.3 nM (TD50 of 5 μg mL^−1^)	[12,21,66,68]
		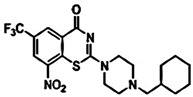 PBTZ169	MIC99 ≤ 0.19 ng mL^−1^; 2.3 nM (TD50 of 58 μg mL^−1^)	[12,21,66]
		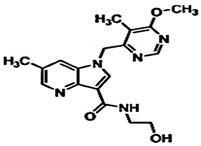 AZ7371	0.64 μg mL^−1^	[18]
		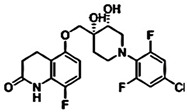 OPC167832	0.24 to 2 ng mL^−1^	[69]
		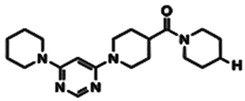 Piperidinylpyrimidines derivatives 3	MIC90 (μM): 35.6	[12]
		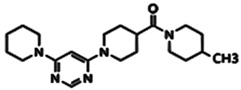 Piperidinylpyrimidines derivatives 4	MIC90 (μM): 15.6	[12]
2.	KasA	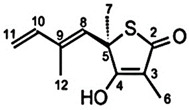 Thiolactomycin (TLM)	IC_50_ n.d.	[32]
		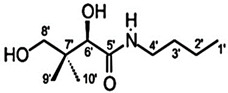 Pantetheine analog, PK940	IC_50_ n.d.	[32]
3.	EthR	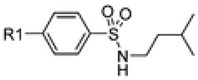 4-Iodo-*N*-prop-2-ynyl benzene sulfonamide (Compound **3**)	SPR IC50 160 µM	[37]
		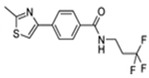 Compound **18** (4-Iodo-*N*-prop-2-ynylbenzenesulfonamide derivative)	IC_50_ 5 µM EC_50_ 6 µM LE 0.34	[37]
		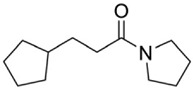 Compound **21** (4-Iodo-N-prop-2-ynylbenzenesulfonamide derivative)	IC_50_280 µM LE 0.36	[33,37]
		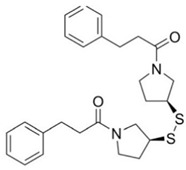 Compound **22** (4-Iodo-N-prop-2-ynylbenzenesulfonamide derivative)	IC_50_ 1 µM LE 0.36	[37]
		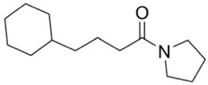 Compound **23** (4-Iodo-N-prop-2-ynylbenzenesulfonamide derivative)	IC_50_ n.d. EC_50_ 0.04 µM	[37]
		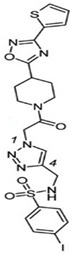 Compound **20** (4-Iodo-N-prop-2-ynylbenzenesulfonamide derivative) 4-Iodo-N-(1-{2-oxo-2-[4-(3-thiophen-2-yl-1,2,4-oxadiazol-5-yl)-piperidin-1-yl]-ethyl}-1H-1,2,3-triazol-4-ylmethyl)-benzenesulfonamide	IC50 = 580 µM	[70]
		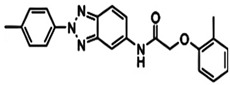 Compound **1** (N-phenyl-149 phenoxy acetamide derivative	IC50 = 2.9 µM	[40]
		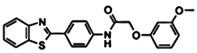 Compound **4** (N-(4-benzothiazol-2-ylphenyl)-2-(3-methoxyphenoxy)acetamide)	EC_50_ 0.21 µM	[40]
		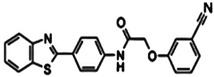 Compound **5** (N-(4-benzothiazol-2-ylphenyl)-2-(3-methoxyphenoxy)acetamide)	EC_50_ 0.34 µM	[40]
4.	7,8-Diaminopelargonic acid (DAPA) synthase (BioA)	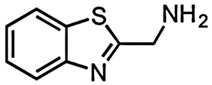 2-(aminomethyl)-benzothiazole 23	IC_50_ n.d.	[56]
		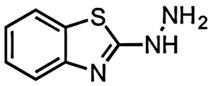 2-(aminomethyl)-benzothiazole 23 hydrazine derivative	IC_50_ n.d.	[56]

n.d.—not determined.

**Table 3 pharmaceuticals-15-01415-t003:** HIV targets and its fragment hit molecules with biological activity.

Target	Analogue	Mode of Action	Reference
HIV-1 Reverse Transcriptase	NBD compound 17	Antiviral activity Cytotoxicity	[73]
RT drug resistant mutants	*p*-Hydroxyaniline 8	Inhibit Rnase H activity	[74]
HIV-1 Integrase	*N*-benzyl indoline 15 and 19, and Y3 compounds	Inhibit HIV-1 IN activity In vitro and In vivo cell culture	[97,100]
	KM-SB3	Inhibit HIV-1 IN activity	[101]
HIV-1 Protease	IF1 and 4D9	Inhibit HIV-1 PR	[105,107]
HIV-1 gp120	NBD-556 and its amine and acid form (NBD-10111)	Antiviral activity Cytotoxicity	[80]
Mutation of HIV-1 gp120 (Arg59 and Asp368) with CD4 receptor	N (2-(2-guanidinoacetamido) -1-(4-(hydroxymethyl) thiazol- 2-yl-ethyl) 54(trifluoromethyl)phenyl)1H-pyrrole-2-carboxamide (Compound **10**)	High antiviral activity and cytotoxicity	[112]
	Compound 882376	Antiviral activity prevents the interaction of gp120 and CD4 Prevents cell to cell fusion	[110]
	NBD-14010 NBD-14171 NBD-14136 NBD-14270 (Thiazole ring NBD compounds)	Moderate antiviral activity	[112]

**Table 4 pharmaceuticals-15-01415-t004:** Structure of HIV targets and its fragment hit molecules.

Structure	Fragment Hit	Reference
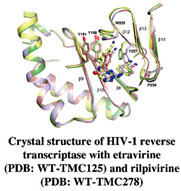	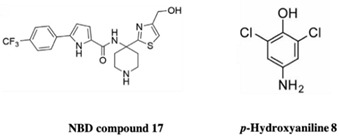	[80,81,118]
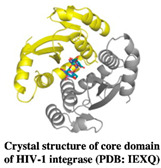	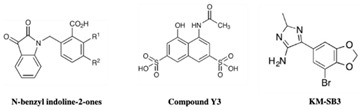	[101]
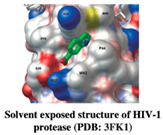	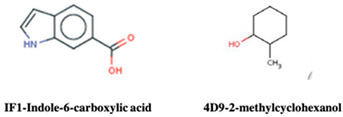	[105]
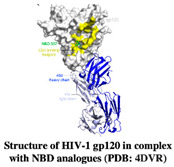	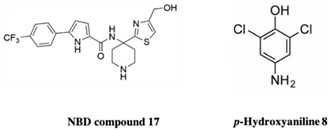	[119]

**Table 5 pharmaceuticals-15-01415-t005:** HIV fragment molecules with their potency parameters.

S.No.	Target	Fragment Molecule	Potency	References
1.	HIV-1 Reverse transcriptase	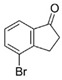 (4-bromo-1-indanone)	IC_50_ < 25 μM	[77]
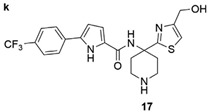 amine piperidine ring fragment 17 NBD compound	1.7 ± 0.2	[80]
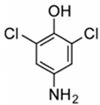 4, 5 (oxime), and 8 (P-hydroxyaniline)	μM range	[81]
2.	HIV-1 Integrase	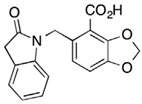 Compound Y3a	IC_50_-259 μM	[97,98]
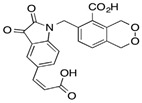 Compound Y3b	IC_50_ 5 μM	[97,98]
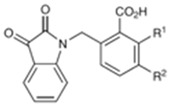 Compound **19** (N-benzyl indolinone analog)	EC_50_ 29 μM	[100]
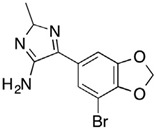 KM-SB3	IC_50_ 170 μM	[101]
3.	HIV-1 Protease	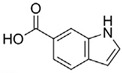 IF1 (derivative of indole-6-propionic acid)	>1 mM	[98]
4.	Gp120	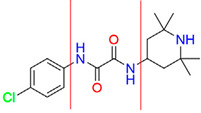 NBD-556 and its amine andacid form (NBD-10111)	IC_50_ n.d.	[80]
Mutation of HIV-1 gp120 (Arg59 and Asp368) with CD4 receptor	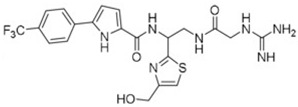 N (2-(2-guanidinoacetamido)-1-(4-(hydroxymethyl) thiazol-2-yl-ethyl) 54(trifluoromethyl)phenyl)1H-pyrrole-2-carboxamide (Compound **10**)	IC_50_ n.d.	[112]
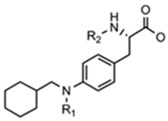 Compound 882376	IC_50_ n.d.	[110]
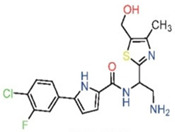 NBD-14010	IC_50_ n.d.	[112]
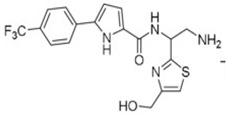 NBD-14171	IC_50_ n.d.	[112]
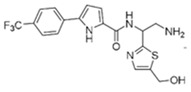 NBD-14136NBD-14270 (Thiazole ring NBD compounds)	IC_50_ n.d.	[112]

n.d.—not determined.

## Data Availability

Data is contained within the article.

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
