# Peer review of "Implications of Fragment-Based Drug Discovery in Tuberculosis and HIV"

_pharmaceuticals, 2022, doi:10.3390/ph15111415_

Round 1

Reviewer 1 Report

 The review title Implications of Fragment-Based Drug Discovery in Tuberculo-sis and HIV focuses on one most challenging research topics in current science. the manuscript has well constructed areas.

however few portions that needs more attentions:

1) make all the chemical structures the appropriate size and the same size in whole manuscript. follow MDPI format.

2) table 1 and table 3; the fragments should follow with potency parameters below the structures.   3) Based on the author's claim, I agree completely that HIV and TB are strongly correlated. and we anticipate a single druggable target would be beneficial for the patients. In particular this discussion is missing in the sections while describing the fragments as well as disease targets. 4) Crystal structure of HIV-1 gp120, this structure should be replaced with a high resolution image and clear visible image.    5) page 9, line 337; 142 kDa of what?   6) The review covers a lot of information but misses some details in few places, like in page 10 line 411; "Later, the binding model was obtained by NMR constraints combined with Au-410 to dock Vina calculations " finally the results of these experiments were not added. 7) page 3, line 28; please mention the parent lead molecule and mention details of the fragment interaction with the target. 8) table 2, STATUS" term meaning more like a clinical stage status, rather can be replaced with modes of action/inhibition of something better.

Author Response

Reviewer comments

Authors response

Reviewer #1:

1.     Make all the chemical structures the appropriate size and the same size in whole manuscript. follow MDPI format.

We thank the reviewers for the comment, we have now maintained the exact size of the chemical structures and pictures in the entire manuscript.

2.     Table 1 and table 3; the fragments should follow with potency parameters below the structures.

We maintain the potency parameters below the structures in table 2 and table 5.

3.     Based on the author's claim, I agree completely that HIV and TB are strongly correlated and we anticipate a single druggable target would be beneficial for the patients. In particular this discussion is missing in the sections while describing the fragments as well as disease targets.

We are thankful to the reviewers for making this point. The present review focuses on progress made in the development of the novel potent FBDD fragments that are directed against druggable TB and HIV targets. Having potent compounds against TB and HIV help in controlling these pathogens as HIV patients are highly susceptible to the development of active TB disease either due to infection or reactivation of latent TB. Identification of host factors that can control both the pathogens TB and HIV will revolutionize the field however so far, no such targets have been identified and no studies on FBDD fragments against such host-directed targets have been performed. Now we have included it (lines 90-96).

4.     Crystal structure of HIV-1 gp120, this structure should be replaced with a high-resolution image and clear visible image.

we thank reviewers for noticing it, the crystal structure of HIV-1 gp120 in the table has been replaced with a clear and high-resolution image (Table 4). 

5.     Page 9, line 337; 142 kDa of what?

We have corrected the sentence (line 436)

6.     The review covers a lot of information but misses some details in few places, like in page 10 line 411; "Later, the binding model was obtained by NMR constraints combined with Au-410 to dock Vina calculations " finally the results of these experiments were not added.

We made changes in the sentences and explained the results clearly for better understanding of the readers (lines 513-517 and 521-539).  

7.     Page 3, line 28; please mention the parent lead molecule and mention details of the fragment interaction with the target.

We included the detailed explanation of parent lead molecule and its interaction with the target in the manuscript.

 4-Iodo-N-prop-2-ynyl benzene sulfonamide (Compound 3) is the parent lead molecule and interacted with the linear ligand binding domain of EthR (lines 156-169).

8.     Table 2, STATUS" term meaning more like a clinical stage status, rather can be replaced with modes of action/inhibition of something better.

As per the reviewer's suggestion, we modified it to “Mode of action” in table 3.

Reviewer 2 Report

Summary

This manuscript summaries the latest developments on tuberculosis and HIV inhibitors developed by FBDD approach. The authors discussed the use of molecules that are identified by FBDD approach to identify novel binding sites on the target and the assays used to evaluate their inhibitory activities. Due to the relevance of this approach towards these targets, this review article would be of interest to the community. However, I would suggested that it can be accepted for publication after some major modifications.

Major issues

line 28: "Mycobacterium tuberculosis that attacks lungs and other parts of the body, and remains the leading cause of death worldwide.

Mycobacterium tuberculosis is a bacteria. The disease caused by this bacteria, can be considered a leading cause of death, but it is not the leading cause (it is ischaemic heart disease and stroke)

line 55: "the fragment-based drug discovery (FBDD), an established futuristic strategy to detect small fragments that bind to a specific target"

Why FBDD is a futuristic approach? The sentence is an apparent paradox because the same approach cannot be simultaneous well established and futuristic.

line 98: In initial screening using SAR, authors have found piperidinylpyrimidines derivatives with a reportable MIC.

What is a reportable MIC? The authors should mention the numeric values and type of MIC (MIC90)

line 130: "The resulting SAR data with 32-fold improved inhibition of EthR DNA binding was found with compound 18. Also, these results were confirmed ex vivo, and they found a boost in the antibacterial effect of ethionamide at 1/10 its MIC

In line with previous comment, MIC values should be included.

Along the manuscript, the authors refer several compounds without referencing their IUPAC name or without depicting its chemical structure. For example: "The authors reported another hit, 21, and it was more interesting because of its ability to bind both the second subpocket in the crystals and the hydrophobic cavity simultaneously. The linking of slightly modified compounds resulted in the formation of compound 22 with a decreased LC50." (line 139). Without the seeing the structures is very hard to track the structural modification. 

Another examples: lines 190, 266, 295-297, 300, 304-305, 376-377, 383, 385-386, 437.

The authors should include the ligands chemical structures named accordingly to the order in which they appear in the text. 

Minor issues

line 56: "Over thirty FBDD compounds are in clinical trials for Cancer (Vemurafenib, kisqali, Balversa and Venetoclax [9]) 57 and TB (BTZ043 [30], PBTZ169 (macozinone) [31], AZ7371 [32], and OPC-167832 [33])."

I understand the reason for this sentence, but at line 76 the authors reference compounds that have been approved. I would suggest to merge this two sentences and they booth validate the relevance of FBDD approach. 

line 282: "Peptide-type inhibitors of LEDGF interaction and multimerization, have also been reported in several studies and represent a new class of non-catalytic site inhibitors (NCINIs) that have entered phase 1 clinical trials"

References are from 2014. These compounds are still in phase 1?

Why the name of drugs is capitalized? (ex: "Isoniazid"). For instance, in line 237/238, nevirapine and efavirenz are not capitalized. This should be normalized, but I would suggest to not capitalize these names.

line 220: "Specifically, several essential enzymes including reverse transcriptase, integrase, protease, gp120, gp41, CCR5, and TAR/Tat are involved in the life cycle of HIV. The blockage of these HIV enzymes prevents virus proliferation."

Please add reference to this sentence.

Specify "Mtb" and "XDR-TB" abbreviations

Check reference "[112]" in Table 1

These words should be in Italics:

- In vivo

- ex vivo

- in vitro

- et al.

- de novo  

- N

- M. smegmatis

- Streptomyces

- Nitrogen atoms in IUPAC names, ex: "N-phenyl"

Delete the white spaces: lines 42, 51, 203, 331, 447

Suggestions

line 71: "Despite the availability of ART therapy against HIV and six-month 4-drug regimen chemotherapy against drug-susceptible Mtb, both the diseases remain at the top list with high mortality rates worldwide. Additionally, the emergence of drug resistance (DR) and multi-drug resistance (MDR) strains demands improved drug treatments. The FBDD approach has been successfully explored on the disease targets by several studies. In cancer, four FBDD drugs have been approved, Vemurafenib, Pexidartinib, Erdafitinib, and Venetoclax."

This information have already been discussion in the introduction. I think that it is reductant.

Format the chemical structures displayed in Table 1. 

Author Response

Reviewer comments

Authors response

Reviewer #2 (Remarks to the Author):

Major points:

1.     line 28: "Mycobacterium tuberculosis that attacks lungs and other parts of the body and remains the leading cause of death worldwide.

Mycobacterium tuberculosis is a bacteria. The disease caused by this bacteria, can be considered a leading cause of death, but it is not the leading cause (it is ischaemic heart disease and stroke)

We thank reviewer for making this point now we have modified our sentence to “TB is one of the major causes of death worldwide leading to 10 million cases and ~1.3 million deaths globally”. (lines 32-33)

2.     Line 55: "the fragment-based drug discovery (FBDD), an established futuristic strategy to detect small fragments that bind to a specific target"

Why FBDD is a futuristic approach? The sentence is an apparent paradox because the same approach cannot be simultaneous well established and futuristic.

Thanks for the comment. We agree that both established and futuristic cannot be simultaneous and the sentence has been modified to “an established strategy” (lines 61-62)

3.     Line 98: In initial screening using SAR, authors have found piperidinylpyrimidines derivatives with a reportable MIC.

What is a reportable MIC? The authors should mention the numeric values and type of MIC (MIC90)

We modified the sentence to “In initial screening using SAR, authors have found piperidinylpyrimidine derivatives with minimum inhibitory concentration (MIC90) of H37Rv was 30.6 μM and 15.6 μM. (lines 124-125)

4.     line 130: "The resulting SAR data with 32-fold improved inhibition of EthR DNA binding was found with compound 18. Also, these results were confirmed ex vivo, and they found a boost in the antibacterial effect of ethionamide at 1/10 its MIC

In line with previous comment, MIC values should be included.

The MIC values are included in the manuscript “(MIC99 μg/ml): BTZ043: 0.001; PBTZ169:0.0003;) (lines 174-175)

5.     Along the manuscript, the authors refer several compounds without referencing their IUPAC name or without depicting its chemical structure. For example: "The authors reported another hit, 21, and it was more interesting because of its ability to bind both the second subpocket in the crystals and the hydrophobic cavity simultaneously. The linking of slightly modified compounds resulted in the formation of compound 22 with a decreased LC50." (line 139). Without the seeing the structures is very hard to track the structural modification.

Another examples: lines 190, 266, 295-297, 300, 304-305, 376-377, 383, 385-386, 437.

The authors should include the ligands chemical structures named accordingly to the order in which they appear in the text.

Thanks for the comment made by the reviewer. We have included the IUPAC names of the chemical structures of the compounds at their respective places in the manuscript.

Minor issues:

6.     Line 56: "Over thirty FBDD compounds are in clinical trials for Cancer (Vemurafenib, kisqali, Balversa and Venetoclax [9]) 57 and TB (BTZ043 [30], PBTZ169 (macozinone) [31], AZ7371 [32], and OPC-167832 [33])."

I understand the reason for this sentence, but at line 76 the authors reference compounds that have been approved. I would suggest to merge this two sentences and they both validate the relevance of FBDD approach. 

The suggested two lines have merged to one sentence. (lines 72-76)

7.     line 282: "Peptide-type inhibitors of LEDGF interaction and multimerization, have also been reported in several studies and represent a new class of non-catalytic site inhibitors (NCINIs) that have entered phase 1 clinical trials"

References are from 2014. These compounds are still in phase 1?

We have included an additional information of NCINIs (Lines 362-374) for better understanding and clarity. We believe that these are in phase 1 clinical trials and there is no available data in recent publication that are entered further clinical pahase.  

8.     Why the name of drugs is capitalized? (Ex: "Isoniazid"). For instance, in line 237/238, nevirapine and efavirenz are not capitalized. This should be normalized, but I would suggest to not capitalize these names.

As per the suggestion, the drug names in the manuscript have normalized.

9.     line 220: "Specifically, several essential enzymes including reverse transcriptase, integrase, protease, gp120, gp41, CCR5, and TAR/Tat are involved in the life cycle of HIV. The blockage of these HIV enzymes prevents virus proliferation."

Please add reference to this sentence.

As per the suggestion, we have added the reference. (line 290)

10.  Specify "Mtb" and "XDR-TB" abbreviations

The suggested abbreviations are included in the manuscript.

11.  Check reference "[112]" in Table 1

We thoroughly checked and inserted the references in table 1

These words should be in Italics:

12.  In vivo

All these words have been changed to italics in the manuscript.

13.  ex vivo

14.  in vitro

15.  et al.

16.  de novo 

17.  N

18.  M. smegmatis

19.  Streptomyces

20.  Nitrogen atoms in IUPAC names, ex: "N-phenyl"

21.  Delete the white spaces: lines 42, 51, 203, 331, 447

The whole manuscript has been thoroughly checked and white spaces in the lines are removed.

Suggestions:

22.  line 71: "Despite the availability of ART therapy against HIV and six-month 4-drug regimen chemotherapy against drug-susceptible Mtb, both the diseases remain at the top list with high mortality rates worldwide. Additionally, the emergence of drug resistance (DR) and multi-drug resistance (MDR) strains demands improved drug treatments. The FBDD approach has been successfully explored on the disease targets by several studies. In cancer, four FBDD drugs have been approved, Vemurafenib, Pexidartinib, Erdafitinib, and Venetoclax."

This information have already been discussion in the introduction. I think that it is reductant.

Thanks for the suggestion, we have removed the repeated lines and edited in the manuscript.

23.  Format the chemical structures displayed in Table 1.

We formatted the chemical structures in table 1

Reviewer 3 Report

The article by Mallakuntla Mohan et al. Implications of Fragment-Based Drug Discovery in Tuberculosis and HIV. In this review the authors emphasize how some potent inhibitors have been developed through an FBDD approach and they also highlight some important targets of mycobacteria and HIV, as well as assays used to assess inhibitory activities. First of all, I would like to thank you for your effort.

This revision requires corrections to improve the quality of the work.

General Recommendations:

1.- In the introduction, it would be advisable to include a schematic representation of the general fragment-based drug discovery (FBDD) method.

2.- In the introduction, it would also be advisable, as an illustration, to place the structures of the compounds mentioned by the authors, that have been successfully obtained through an FBDD approach or those that at some point in its development used this method.

3.- It is also recommended that at the end of each section (for each specific target) a schematic representation of the optimization of the compounds successfully obtained by means of FBDD is placed, as well as the activity values. This in order to be able to appreciate the possible modifications and design of the compound successfully obtained by this technique.

4.- In those cases where the development of a new compound involves a pharmacophoric model, a schematic representation of this model should be placed and discussed in order to appreciate the design and the new binding mode of the compound.

5.- Table 1 and Table 3 need to be improved. chemical structures must be drawn properly, where they all have the same size and shape, that is, the same object setting must be applied for each structure, not copied and pasted from another source.

6.- In the tables Table 1 and Table 3, the images of the molecular docking representing the interactions of the different fragments should keep the same size and if possible the same style. As well as the PDB codes of each crystalline structure used must be reported below each image.

Author Response

Reviewer comments

Authors response

Reviewer #3

The article by Mallakuntla Mohan et al. Implications of Fragment-Based Drug Discovery in Tuberculosis and HIV. In this review the authors emphasize how some potent inhibitors have been developed through an FBDD approach and they also highlight some important targets of mycobacteria and HIV, as well as assays used to assess inhibitory activities. First of all, I would like to thank you for your effort.

This revision requires corrections to improve the quality of the work.

General Recommendations:

1.     In the introduction, it would be advisable to include a schematic representation of the general fragment-based drug discovery (FBDD) method.

We inserted the schematic representation of FBDD methodology as figure 2.

2.     In the introduction, it would also be advisable, as an illustration, to place the structures of the compounds mentioned by the authors, that have been successfully obtained through an FBDD approach or those that at some point in its development used this method.

Thanks for the suggestion. We inserted the illustration of FBDD inhibitors are in clinical trials as figure 1.

3.     It is also recommended that at the end of each section (for each specific target) a schematic representation of the optimization of the compounds successfully obtained by means of FBDD is placed, as well as the activity values. This in order to be able to appreciate the possible modifications and design of the compound successfully obtained by this technique.

We thank reviewer for this suggestion. We have mentioned target hit molecules with their potency parameters in table 2 and 5. As the focus of this review is to highlight the novel hit-fragments identified against Mtb and HIV targets using FBDD approach. Many of the hit- fragments against cancer are successfully identified using FBDD approach are in clinical trials.

4.     In those cases where the development of a new compound involves a pharmacophoric model, a schematic representation of this model should be placed and discussed in order to appreciate the design and the new binding mode of the compound.

We thank the reviewers for the illustrative query. However, explaining the whole pharmacophore model for the single target molecule will deviate from the moto of reviewing all targets in a single article. As we have already, we have cited the original research and explained fragment-based hit-to-lead optimization leading to the generation of potent compound 18

5.     Table 1 and Table 3 need to be improved. chemical structures must be drawn properly, where they all have the same size and shape, that is, the same object setting must be applied for each structure, not copied, and pasted from another source.

In the tables Table 1 and Table 3, the images of the molecular docking representing the interactions of the different fragments should keep the same size and if possible, the same style. As well as the PDB codes of each crystalline structure used must be reported below each image

We modified the table 1 and 4 (table no. changed) by maintaining same size and shape of the clear pictures, along with PDB codes.

Round 2

Reviewer 1 Report

the queries are comprehensively addressed and incorporated.

Reviewer 2 Report

I consider that after the revisions made by the authors, the paper now meets the quality requirements for publication.